# DeepHiC: A generative adversarial network for enhancing Hi-C data resolution

**Hao Hong** [1,☯], **Shuai Jiang** [1,☯], **Hao Li** [1], **Guifang Du** [1], **Yu Sun** [1], **Huan Tao** [1], **Cheng Quan** [1], **Chenghui Zhao** [1], **Ruijiang Li** [1], **Wanying Li** [1], **Xiaoyao Yin** [2], **Yangchen Huang** [2], **Cheng Li** [3,4]*, **Hebing Chen** [1]*, **Xiaochen Bo** [1]*

**1** Beijing Institute of Radiation Medicine, Beijing, China, **2** College of Computer, National University of Defence Technology, Changsha, China, **3** Peking-Tsinghua Center for Life Sciences, Academy for Advanced Interdisciplinary Studies; School of Life Sciences, Peking University, Bejing, China, **4** Center for Statistical Science, Center for Bioinformatics, Peking University, Beijing, China

☯ These authors contributed equally to this work.

\* cheng_li@pku.edu.cn (CL); chb-1012@163.com (HC); boxc@bmi.ac.cn (XB)

## Abstract

Hi-C is commonly used to study three-dimensional genome organization. However, due to the high sequencing cost and technical constraints, the resolution of most Hi-C datasets is coarse, resulting in a loss of information and biological interpretability. Here we develop DeepHiC, a generative adversarial network, to predict high-resolution Hi-C contact maps from low-coverage sequencing data. We demonstrated that DeepHiC is capable of reproducing high-resolution Hi-C data from as few as 1% downsampled reads. Empowered by adversarial training, our method can restore fine-grained details similar to those in high-resolution Hi-C matrices, boosting accuracy in chromatin loops identification and TADs detection, and outperforms the state-of-the-art methods in accuracy of prediction. Finally, application of DeepHiC to Hi-C data on mouse embryonic development can facilitate chromatin loop detection. We develop a web-based tool (DeepHiC, http://sysomics.com/deephic) that allows researchers to enhance their own Hi-C data with just a few clicks.

## Author summary

We developed a novel method, DeepHiC, for enhancing Hi-C data resolution from low-coverage sequencing data using generative adversarial network. DeepHiC is capable of reproducing high-resolution (10-kb) Hi-C data with high quality even using 1/100 down-sampled reads. Our method outperforms the previous methods in Hi-C data resolution enhancement, boosting accuracy in chromatin loops identification and TADs detection. Application of DeepHiC on mouse embryonic development data shows that enhancements afforded by DeepHiC facilitates the chromatin loops prediction of these data, yielding significant interactions more enriched in open chromatin regions and gene promoters. We also developed a user-friendly web server (http://sysomics.com/deephic) that allows researchers to enhance their own low-coverage Hi-C data with just few clicks.

**Data Availability Statement:** All Hi-C data are available from the GEO database (accession numbers GSE63525, GSE82185). ChIA-PET data are available from the ENCODE (accession number ENCSR000CAC). DNase-seq data are available

**Funding:** This work was funded by rewards including the National Natural Science Foundation of China (No. 31801112), the Beijing Nova Program of Science and Technology (NO. Z191100001119064), URL: http://www.nsfc.gov. cn and https://mis.kw.beijing.gov.cn, to HC and the National Natural Science Foundation of China (No. 61873276), URL: http://www.nsfc.gov.cn, to XB. The funders had no role in study design, data collection and analysis, decision to publish, or preparation of the manuscript.

**Competing interests:** The authors have declared that no competing interests exist.

# Introduction

The high-throughput chromosome conformation capture (Hi-C) technique [1] is a genome-wide technique used to investigate three-dimensional (3D) chromatin conformation inside the nucleus. It has facilitated the identification and characterization of multiple structural elements, such as the A/B compartments [1], topological associating domains (TADs) [2, 3], enhancer-promoter interactions [4] and stripes [5] over recent decades. In practice, Hi-C data is conventionally stored as a pairwise read count matrix $M_{n \times n}$, where $M_{ij}$ is the number of observed interactions (read-pair count) between genomic regions $i$ and $j$, and the genome is partitioned into $n$ fixed-size bins (e.g., 25 kb). Bin size (i.e., resolution), is a crucial parameter for Hi-C data analysis, as it directly affects the results of downstream analysis, such as predictions of enhancer-promoter interactions [6–11] or identification of TAD boundaries [6, 12–16]. Depending on sequencing depths, the size of commonly used bins ranges from 1 kb to 1 Mb.

Because of the high cost of sequencing, most available Hi-C datasets have relatively low resolution, such as 25 kb or 40 kb [17]. Sequencing high-resolution Hi-C matrices demands sufficient sequencing coverage; otherwise, the contact matrix would be extremely sparse and contain excessive stochastic noise. When sequencing Hi-C data, billions of read-pairs are typically necessary to achieve truly genome-scale coverage at kilobase-pair resolution [18], and the cost of Hi-C experiments generally scales quadratically with the desired level of resolution [19]. Low-resolution data may be sufficient for detecting large-scale genomic patterns such as A/B compartments, but the decrease in resolution when analyzing Hi-C data may prevent identification of fine-scale genomic elements such as sub-TADs [20, 21] and enhancer-promoter interactions, and even lead to inconsistent results when detecting interactions and TADs in replicated samples [22]. Therefore, developing a computational model to impute a higher-resolution Hi-C contact matrix from currently available Hi-C datasets show its potency and usefulness.

Several pioneering works on solving problems related to low-resolution Hi-C data have recently emerged. Li *et al.* proposed deDoc for detecting megabase-size TAD-like domains in ultra-low resolution Hi-C data [23]. Zhang *et al.* proposed a deep learning model called HiCPlus to enhance Hi-C matrices from low-resolution Hi-C data [17]. HiCPlus showed that chromatin interactions can be predicted from their neighboring regions, by using the convolutional neural network (CNN) [24]. Carron *et al.* proposed a computational method called Boost-HiC for boosting reads counts of long-range contacts [25]. And Liu *et al.* proposed HiCNN [26] which is a 54-layer CNN and achieved better performance than HiCPlus. While these results were encouraging, three problems still exist in Hi-C data resolution enhancement algorithms. First, Hi-C data contain numerous high-frequency details ($M_{ij}$ and its nearby values are very large, while values in neighboring regions are small) and sharp edges, which are usually considered to indicate the presence of enhancer-promoter loops, stripes, and TAD boundaries. Models that rely on regression and mean squared error (MSE) loss, which is thought to yield solutions with overly smooth textures [27], are likely to smooth these features. Thus, we seek to develop a model which is capable of predicting data with a sharp or degenerated distribution. Second, the structural patterns and textures of Hi-C data are abundant. The hypothesis space, which is controlled by the number of parameters, should be able to capture richer structures as it grows [28]. It is possible that increasing the depth of network would increase accuracy [29], while ensuring the model's generalizability and restraining the overfitting problem. The final critical problem is the stochastic noise in Hi-C data. An effective model should be able to predict solutions resides on the manifold of target data and thus diminish stochastic noise (i.e., capability for denoising) [30, 31].

In order to make accurate prediction of high-resolution Hi-C data from low-coverage sequencing samples against these three problems. We developed a deep learning model which employed the state-of-the-art generative adversarial network (GAN), in combination with some advanced techniques in deep learning field. Goodfellow *et al.* first introduced the GAN model for estimating generative models with an adversarial process [32]. The GAN architecture allows the generative net to easily learn target data distribution, even sharp or degenerated distribution. GAN has been used for various applications and is showing its huge potency. For instance, Mirza *et al.* proposed the conditional GAN (cGAN) of which the generator learns the data distribution upon conditional inputs [33]. Li and Wand described the usage of GANs to learn a mapping from one manifold to another [34]. Another inspiring work for us was described by Ledig *et al.* [35], who proposed SRGAN to generate photo-realistic super-resolution images. Besides, He *et al.* introduced the concept of residual learning and proved that an ultra-deep neural network could be easily trained via residual learning and achieve superior performance [36]. Also, researchers started to design task-specified loss functions, using not only MSE loss (i.e., L2 loss) but other losses like perceptual loss [37] as well, and gain surprising advancements [38].

In this paper, we propose a GAN-based method DeepHiC to enhance the resolution of Hi-C data. Using low-coverage Hi-C matrices (obtained by downsampling original Hi-C reads) as input, we demonstrate that DeepHiC is capable of reproducing high-resolution Hi-C matrices. DeepHiC-enhanced data achieve high correlation and structure similarity index (SSIM) compared with original high-resolution Hi-C matrices. And even using as few as 1% original reads, while no previous methods enhancing data of this depth, DeepHiC is still capable of inferring high-resolution data and achieves the correlation and SSIM score as good as the real high-resolution replicated assay. Compared with previous methods, our method is more accurate in predicting high-resolution Hi-C data, even in fine-grained details, and performed better when applied to different cell lines. Enhancements of DeepHiC improve the accuracy of downstream analysis such as identification of chromatin loops and detection of TADs. In this study, we applied DeepHiC to Hi-C data in mouse embryonic development and demonstrated that, compared with the original low-coverage Hi-C data, DeepHiC-enhanced Hi-C data enables the identification for chromatin loops that are similar to those identified in deeply sequenced Hi-C data. Besides, we also develop a web-based tool (DeepHiC, http://sysomics.com/deephic) that allows researchers to enhance their own Hi-C data with just a few clicks. In summary, this work introduces an effective model for enhancing Hi-C data resolution and establishes a new framework for prediction of a high-resolution Hi-C matrix from low-coverage data.

## Results

### Parameters training of DeepHiC model

In the current study, we propose a conditional generative adversarial network (cGAN), DeepHiC, for enhancing Hi-C data from low-resolution samples. It contains a generative network *G* and a discriminative network *D*. The former takes low-resolution data as input and imputes the enhanced output, while the latter is only employed during training process as a discriminator for reporting the differences between enhanced outputs and real high-resolution Hi-C data to the network *G*, which form the adversarial training (Fig 1A). Also, in order to alleviate the overly-smooth problem caused by MSE loss, we utilized the perceptual loss to capture structure features in Hi-C contact maps and the total variation (TV) loss for suppressing artifacts [39]. The detailed architecture of DeepHiC is depicted in S1 Fig. The GAN framework benefits *G* network by efficiently capturing the distribution of target data (even very sharp or degenerate distributions) [32] and favors solutions residing on the manifold of target data.

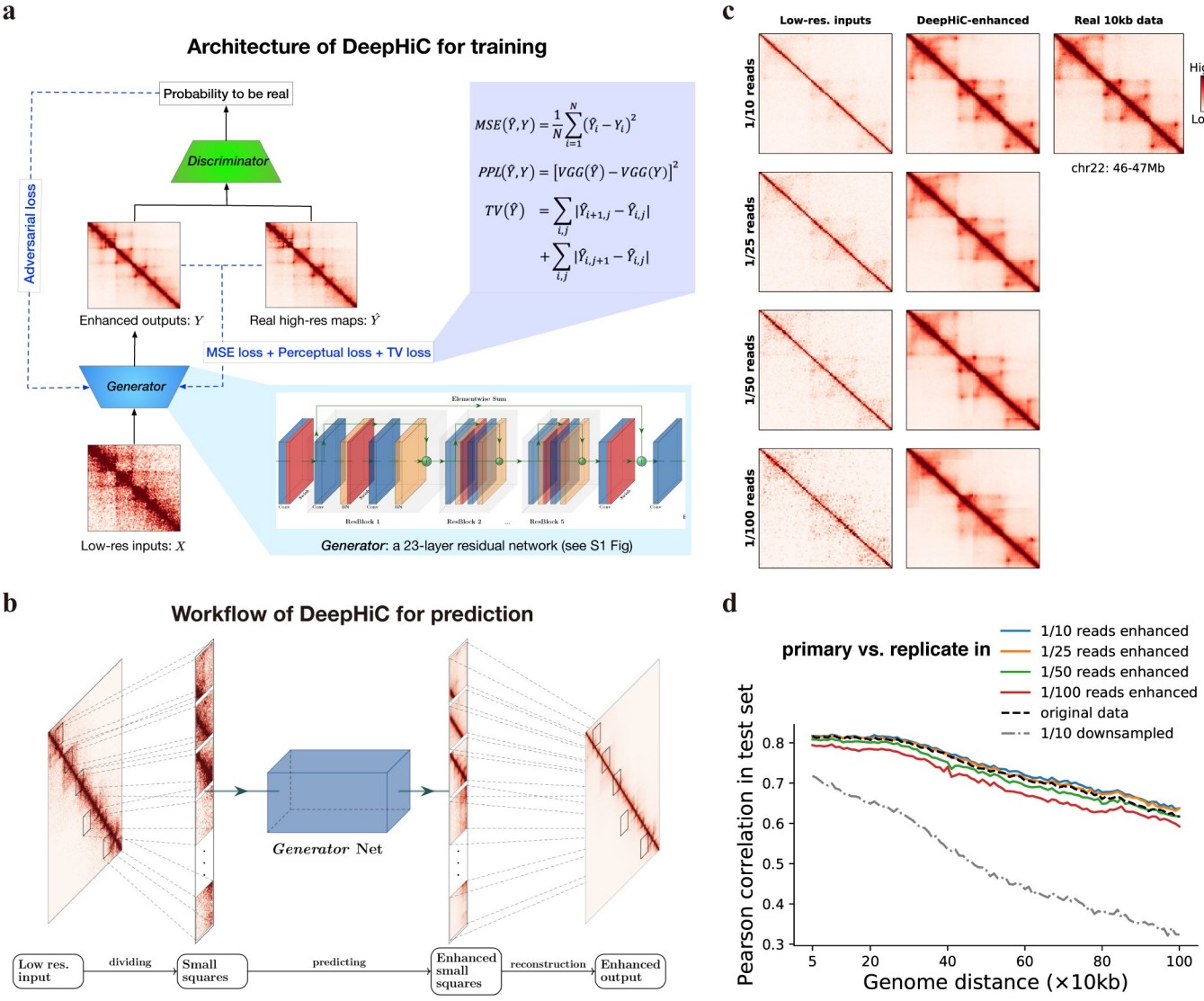

**Fig 1. Overview of the DeepHiC. (a)** DeepHiC framework: low-resolution inputs are obtained by randomly downsampling original reads. It imputes enhanced contact maps using a 23-layer residual network called *Generator*. In the training process, the enhanced outputs are approaching real high-resolution matrices by minimizing mean square error (MSE) loss, perceptual loss (PPL), and total variation (TV) loss, meanwhile, a *Discriminator* network distinguishes enhanced outputs from the real ones and reports the probabilities of enhanced outputs to be real to the *Generator* through adversarial (AD) loss. The imputation and discrimination steps form the adversarial training process. **(b)** For prediction, a low-resolution Hi-C matrix is divided into small squares as inputs. Then enhanced small squares are predicted by the *Generator*. Finally, those squares are merged into a chromosome-wide contact map as the enhanced output. **(c, d)** We randomly downsampled the original reads (obtained from GEO GSE63525) to 1/10, 1/25, 1/50, and 1/100 reads to simulate low-resolution inputs. DeepHiC is trained on chromosomes 1–14 and tested on chromosomes 15–22 (i.e., test set), in GM12878 cell line. **(c)** The trained DeepHiC model can be used for enhancing low-coverage sequencing Hi-C data, as an example which shows a 1Mb-width sub-region on chromosome 22 and **(d)** obtain high correlations between DeepHiC-enhanced matrices and real high-resolution Hi-C at each genomic distance. Colorbar setting: see S1 Note.

We trained DeepHiC on chromosomes 1–14 and tested on chromosomes 15–22 in the *GM12878* cell line dataset during the training process (see Materials and Methods: Implementation of DeepHiC and evaluation). For low-resolution (low-coverage) data with different downsampling ratios, we obtained their corresponding trained models separately. We evaluated the structure similarity index (SSIM) scores in the test set during training. Higher SSIM scores between enhanced output and real high-resolution Hi-C indicate greater structural similarity. For low-resolution data from different downsampling ratios, SSIM scores in the test set

increased gradually and converged when DeepHiC was trained in 200 epochs (S2 Fig), as well as another metric related to MSE (S3 Fig). Generator loss in both the training and test sets decreased simultaneously during the training process (S4 Fig). These results indicate that the model converged successfully in training without overfitting. Furthermore, we tested various splits of training and test sets, like a 5-fold cross validation. Performances in the test set were consistent across different dataset splits, showing that our model is capable of capturing common information from the different training sets and its parameters could be stably derived with no relation to training/test set we used (S5 Fig). We also trained the *generator* net as a regression model without the adversarial part, but SSIM scores in the test set vibrated substantially (S6 Fig). These results suggest that the GAN-based framework efficiently restrains the over-fitting phenomenon and its necessity for prediction. Besides, DeepHiC is also could be trained in *IMR90* or *K562* dataset (S7 Fig).

In the prediction step, we divided the large Hi-C matrix into small squares as model inputs. For a fair comparison in the following analysis, we divided the low-resolution Hi-C matrix into 0.4 Mb × 0.4 Mb sub-regions (40 × 40 bins in 10-kb resolution) same with what HiCPlus does. Then the completed enhanced Hi-C matrix could be obtained by reconstructing all enhanced sub-regions after prediction (Fig 1B) (see Methods: dividing and reconstructing matrices).

## DeepHiC reproduces high-resolution Hi-C from as few as 1% downsampled reads

We used the high-resolution Hi-C data in the GM12878, K562 and IMR90 cell lines from Rao's Hi-C (access code GSE63525) in our experiments. Datasets pertaining to different cell types are denoted as *GM12878*, *GM12878R*, *K562*, and *IMR90* for convenience (*GM12878R* represents the replicated assay in the GM12878 cell line). First, we constructed high-resolution (10-kb) contact matrices using all the reads from the raw data. Then we downsampled the reads to different ratios (ranges from 1:10 to 1:100) of the original reads to simulate the low-resolution Hi-C data. We also constructed contact matrices at the same bin size. Therefore, we obtained paired high-resolution and low-coverage Hi-C data (both were binned at 10-kb). The original experimental high-resolution data were regarded as ground truth in the following analysis, while the low-coverage data were enhanced by DeepHiC using the trained model.

Fig 1C shows the model's enhancements in a 1Mb sub-region (100 bins) on chromosome 22 in the test set (GM12878 cell line). Comparing with the real 10-kb Hi-C data, DeepHiC-enhanced matrices recover patterns such as chromatin loops and TADs successfully from low-coverage inputs. Quantitatively, DeepHiC-enhanced data achieve the correlations as good as the experimental replicate (i.e., *GM12878R*), even though they were predicted from 1% downsampled data (Fig 1D). It also shows the same result in SSIM measure (S8 Fig). These results indicate that the DeepHiC model is capable of reproducing high-resolution Hi-C data with high similarity even using 1% downsampled reads. Because the high-resolution data we used is at 10-kb resolution, it implies that our method could enhance 1Mb resolution Hi-C data to 10-kb resolution with high quality. And there is no available imputation algorithm for enhancing Hi-C data from such a sequencing depth before.

In the following analyses, we trained DeepHiC in 1/16 downsampled data for fairly comparing with other baseline methods such as HiCPlus, Boost-HiC, and HiCNN (see Methods). The trained model we used was trained on data of chromosome 1–14 in *GM12878* dataset. SSIM scores converged at 0.9 in remaining chromosomes (S9 Fig). For remaining chromosomes' data in *GM12878* dataset, as well as the whole *GM12878R*, *K562*, and *IMR90* datasets, we applied the trained model to their downsampled data, then evaluated the performance with taking the real high-resolution data as the ground truth.

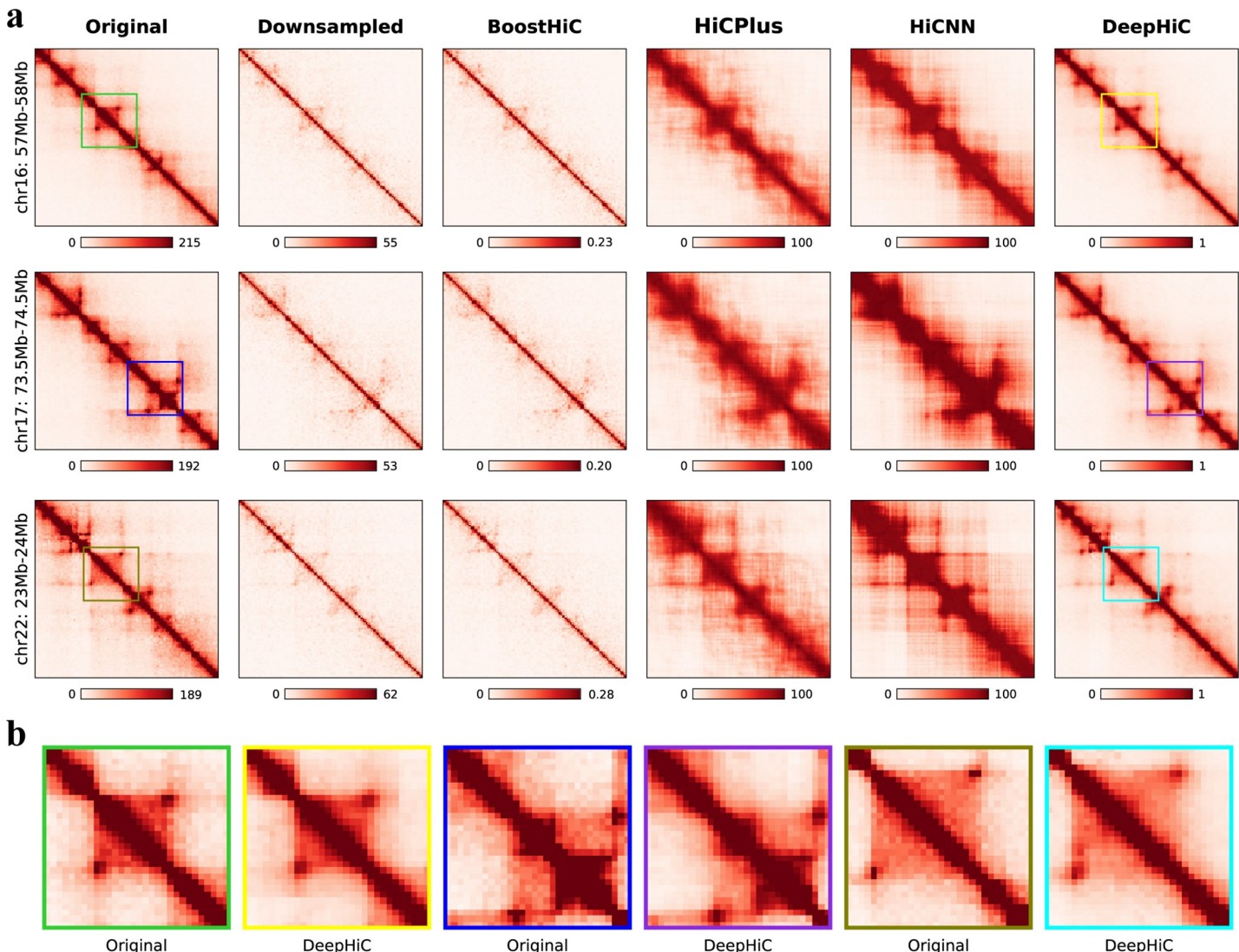

**Fig 2. DeepHiC enhances the interaction matrix, even in fine-grained textures, with low-sequence depth. (a)** Shown in the figures are real (first column), 1/16 downsampled (second column), Boost-HiC/HiCPlus/HiCNN-enhanced (third-fifth columns) and DeepHiC-enhanced (sixth column) interaction matrices in three different 1-Mb-width sub-regions from the GM12878 cell line at 10-kb resolution. **(b)** Enlarged heatmaps of smaller sub-regions (0.3Mb×0.3Mb, extracted from the matching coloured frames in **(a)** obtained from real high-resolution and DeepHiC-enhanced matrices.

## Enhancements of low-resolution data

We first investigate the enhancements afforded by DeepHiC by visualizing data in the form of heatmaps (see S1 Note for colorbar settings). Fig 2A shows three 1-Mb-width sub-regions (arranged by rows) on chromosomes 16, 17, and 22 which were extracted from the test set in the *GM12878* dataset. The real high-resolution examples marked as "Original" in the first column contain clear individual chromatin loops and TAD structures, while low-coverage examples marked as "Downsampled" (second column) have abundant noise and less clear TAD structures. We found that DeepHiC-enhanced data (last column) could accurately restore the patterns and textures which are exactly the same as those in real high-resolution data. Baseline models' results are shown in the third to fifth columns. Noting that Boost-HiC was specifically developed for enhancing long-range contacts [25]. So, it makes sense that Boost-HiC has slight

changes in short-range contacts (third column). The HiCPlus-enhanced data marked as "HiC-Plus" (fifth column) contains much less noise and more visible TAD structures, but refined structures such as chromatin loops are replaced by smooth textures. So does the HiCNN (fifth column), which is a deeper CNN and relies on MSE loss as well. In terms of fine-grained details, we scrutinized smaller 0.3 Mb × 0.3 Mb (30 × 30 bins) sub-regions from these three examples in real high-resolution Hi-C and DeepHiC-enhanced Hi-C, as illustrated in Fig 2B. High similarity between experimental high-resolution data and DeepHiC-enhanced data was observed. Sharp edges in heatmaps, which are deemed difficult to recover in practice, were accurately recovered by DeepHiC. We also visualized three sub-regions from the *GM12878R* dataset (S10 Fig), three sub-regions from the *K562* dataset (S11 Fig), and three sub-regions from the *IMR90* dataset (S12 Fig). And DeepHiC outperforms baseline models in all four data-sets. The SSIM scores for downsampled, HiCPlus-enhanced, HiCNN-enhanced, and DeepHiC-enhanced data, as compared with real high-resolution data for these three sub-regions were 0.20, 0.64, 0.59, and 0.89 on average, respectively.

## DeepHiC outperformed other methods in terms of genome-wide similarity to ground truth deeply sequenced data

Furthermore, we quantitatively investigated genome-wide performance for all four datasets. We calculated SSIM scores for downsampled and various model-enhanced data, as compared with real high-resolution data for all 1 Mb × 1 Mb (100 × 100 bins) sub-regions with non-over-lap at the diagonal across the entire genome (S13 Fig). Fig 3A shows that DeepHiC-enhanced matrices had the highest SSIM scores for all 23 chromosomes in the *GM12878* dataset. Average values for downsampled, HiCPlus-enhanced, HiCNN-enhanced, and DeepHiC-enhanced data were 0.15, 0.71, 0.66, and 0.89, respectively. SSIM scores derived from DeepHiC, HiCPlus, and HiCNN are denoted as $SSIM_{deephic}$, $SSIM_{hicplus}$ and $SSIM_{hicnn}$, respectively. Fig 3B shows the differences between these scores for all 4 datasets covering all chromosomes. Their absolute values are shown in S14 Fig. The comparison results show that DeepHiC achieves greater simi-larity than HiCPlus and HiCNN.

We also computed the Pearson correlation coefficients between the experimental high-res-olution, downsampled, baselines-enhanced, and DeepHiC-enhanced matrices at each genomic distance, which also performed in previous studies. As shown in Fig 3C, the DeepHiC-enhanced matrices obtained higher correlation coefficients (~5%) than the HiCPlus-enhanced matrices at all genomic distances of interest from 50 kb to 1 Mb. This region included proxi-mal and distal regions. We also computed the differences between correlations derived from DeepHiC with those derived from HiCPlus/HiCNN, which are denoted as $r_{deephic}$ and $r_{hicplus}$/$r_{hicnn}$, respectively. Then we investigated the distribution of differences in all four datasets by boxplots, with extremely small p-values obtained for that $r_{deephic}$ are significantly higher than $r_{hicplus}$/$r_{hicnn}$ (paired t-test, pair number = 96), as shown in Fig 3D. Their absolute values are shown in S15 Fig. The results of similarity and correlation comparison revealed our model's advantages in restoring high-resolution Hi-C. More importantly, advantages across various cell lines revealed that DeepHiC can be used to enhance the Hi-C matrix for other cell types.

We omitted comparison with Boost-HiC considering that it aims to enhance long-range contacts. Evaluation of Boost-HiC is plotted in S14 Fig and S15 Fig. Besides, we also investi-gated the performance of detecting A/B compartments for DeepHiC and Boost-HiC, because the latter is reported for it. S16 Fig shows our model achieves comparative performance in detecting A/B compartments, considering our model is trained using short-range contacts.

Besides, we applied DeepHiC to data from various downsampled ratios (e.g., 1/25, 1/36), while still using the trained model derived from 1/16 downsampled data. S17 Fig shows that

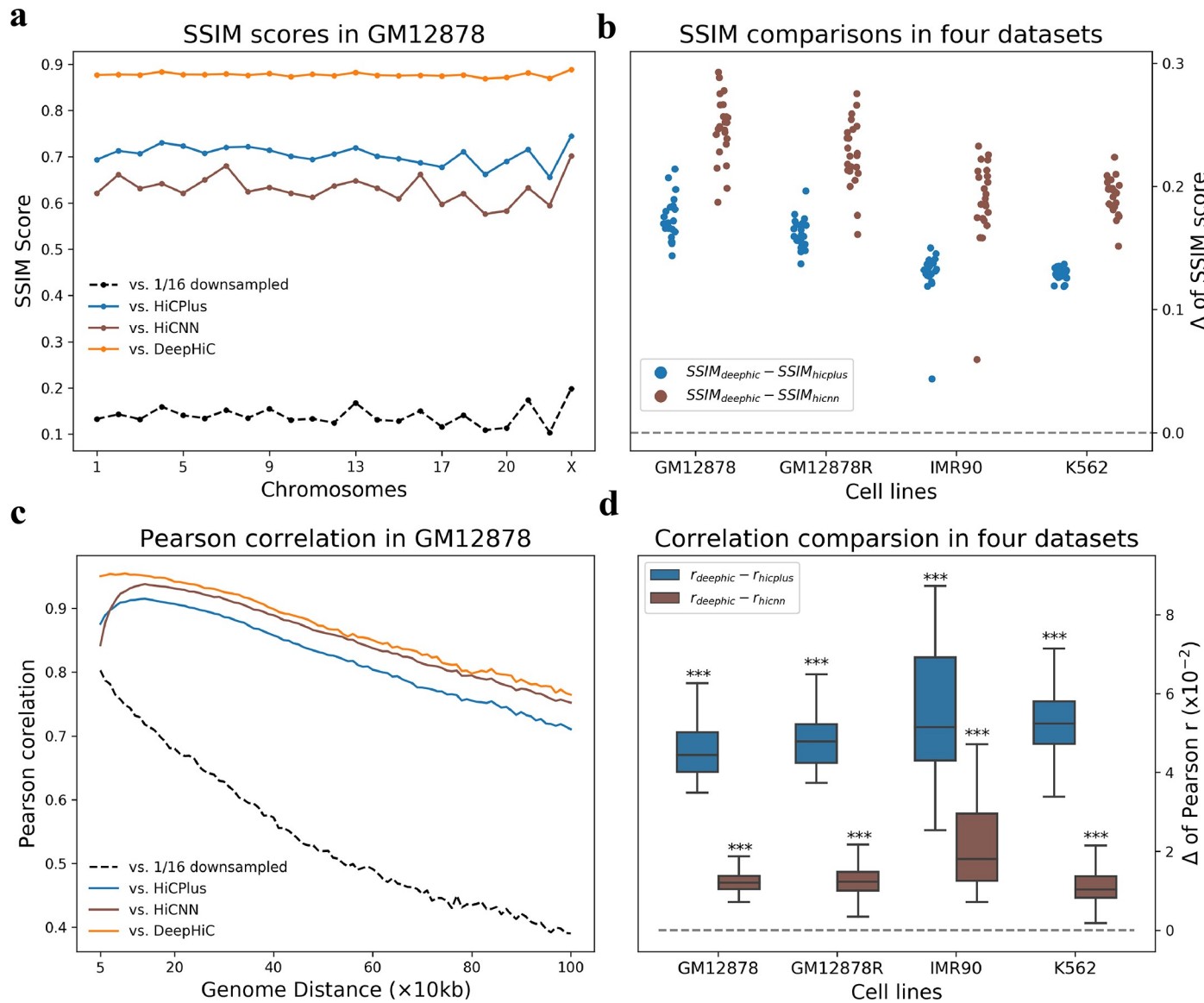

**Fig 3. Genome-wide comparative analyses of similarity and correlation in various cell types. (a)** High SSIM scores between DeepHiC-enhanced and real high-resolution matrices for all chromosomes in the *GM12878* dataset. **(b)** In extending this analysis to other cell lines, we calculated the differences SSIM scores derived from DeepHiC and baseline models. Circle dots represent the Δ values on each chromosome. Dotted line represents the location of zero value. **(c)** Comparison of Pearson correlation coefficients between non-experimental data and real Hi-C data at each genomic distance of interest from 50kb to 1Mb. DeepHiC outperforms other methods at all genomic distances examined. **(d)** We calculated all differences (Δ) between correlations derived from DeepHiC and those derived from HiCPlus/HiCNN at each distance in four datasets. The results obtained are depicted with boxplots. All Δ values are significantly greater than zero (dotted line) (paired t-test, pair number = 96). The whiskers are 5 and 95 percentiles. ***: p-value $< 1\times10^{-20}$.

DeepHiC still achieves greater correlation coefficients. These results suggest that DeepHiC could be employed to enhance low-coverage sequencing data, rather than just enhancing data with a particular ratio. Thus, we used the same downsampling and predicting procedure to make predict on more cell types' data (including mouse cell line CH12-LX) from Rao et al [4], as shown in S18 Fig. And correlations across cell types suggest that our model also preserve the specificities between cell lines (S18D Fig). Further, performances on Hi-C data prepared using 6-cutter enzyme revealed that our model is also applicable to 6-cutter enzyme prepared Hi-C data (S19 Fig).

## Significant interactions in high-resolution Hi-C were accurately recovered from DeepHiC-enhanced matrices

After demonstrating that DeepHiC can restore high-resolution Hi-C from low-resolution data, we investigated whether these enhanced high-resolution matrices could facilitate the identification of significant chromatin interactions, which are usually considered to be chromatin loops. For this purpose, we used Fit-Hi-C software to obtain significant intra-chromosomal interactions. We applied Fit-Hi-C to Hi-C data present above, in four datasets, using the same parameters (Methods). Statistical confidence values (i.e., q-values) for all loci-pairs were acquired by Fit-Hi-C. We kept the predicted significant interactions (q-value $< 1 \times 10^{-6}$) for genomic distances from 20 kb to 1 Mb for further comparative analysis. At first, we visualized three 1 Mb-wide sub-regions. Significant interactions are presented in yellow in the upper triangles of heatmaps (Fig 4A). Compared with the real high-resolution data, only DeepHiC-enhanced matrices yield consistent results in recognizing significant interactions. And the yellow-marked anchors are indeed significant interactions by observing the lower triangular parts of heatmaps. The numbers of interactions in these three sub-regions (denoted as I, II and III) derived from various contact matrices are presented in S20 Fig. HiCNN and HiCPlus-enhanced matrices identified few loci-pairs, while the experimental and DeepHiC-enhanced matrices identified about 40 loci-pairs, respectively. Fig 4A presents the significant interactions identified in real high-resolution Hi-C gathered in 8, 20, and 11 clusters, respectively. However, for low-resolution Hi-C, few interactions were identified. For HiCPlus-enhanced Hi-C, only six clusters were recovered. Surprisingly, DeepHiC-enhanced Hi-C recovered nearly all clusters (35 in total) and no false-positive cluster was added.

Because Fit-Hi-C calculated the significance of all loci-pairs within the genomic distance of interest, we performed a genome-wide comparative analysis by analyzing the significance matrices formed with q-values. We calculated the similarity of significance matrices, as previously performed for Hi-C matrices. Fig 4B shows the Pearson correlation coefficients for significance matrices in the *GM12878* dataset at each genomic distance. Same results of comparisons between the other three datasets are presented in S21 Fig. We observed that q-values derived from DeepHiC-enhanced data were more similar to the real high-resolution data than any others for the entire dataset. We also compared the overlap of identified interactions with real high-resolution data at each genomic distance, as shown in Fig 4C. The Jaccard index (*JI*) of identified interactions between DeepHiC-enhanced data and real high-resolution data was higher at each genomic distance. In addition to using the aforementioned threshold for q-values, we tried more thresholds by scanning various false discovery rates (FDR), ranging from 0.001 to 0.05, with step size of 0.001. We evaluated the overlap of identified interactions according to FDR scanning. We found that DeepHiC outperformed others (Fig 4D). These results suggested that DeepHiC-enhanced Hi-C data are more accurate in predicting chromatin loops and yield less artifact noise.

Next, we compared the significant interactions identified in these Hi-C matrices with the identified chromatin interactions by CTCF chromatin interaction analysis by paired-end tagging sequencing (ChIA-PET) in the K562 cell line, for which related data is available in the ENCODE project. ROC analysis was performed in the same way as is described in HiCPlus, using the identified CTCF-mediated chromatin interactions from ChIA-PET as true positives. As for negatives, we randomly selected the same number of loci pairs that were not predicted to be interacting pairs by ChIA-PET and that had the same distance distribution with positives (10 repeats). We then plotted the ROC (receiver operating characteristic) curve and calculated the area under the ROC curve (AUC) for each. As shown in Fig 4E, CTCF interacting pairs and non-interacting pairs were separated from the DeepHiC-enhanced matrix in the predicted

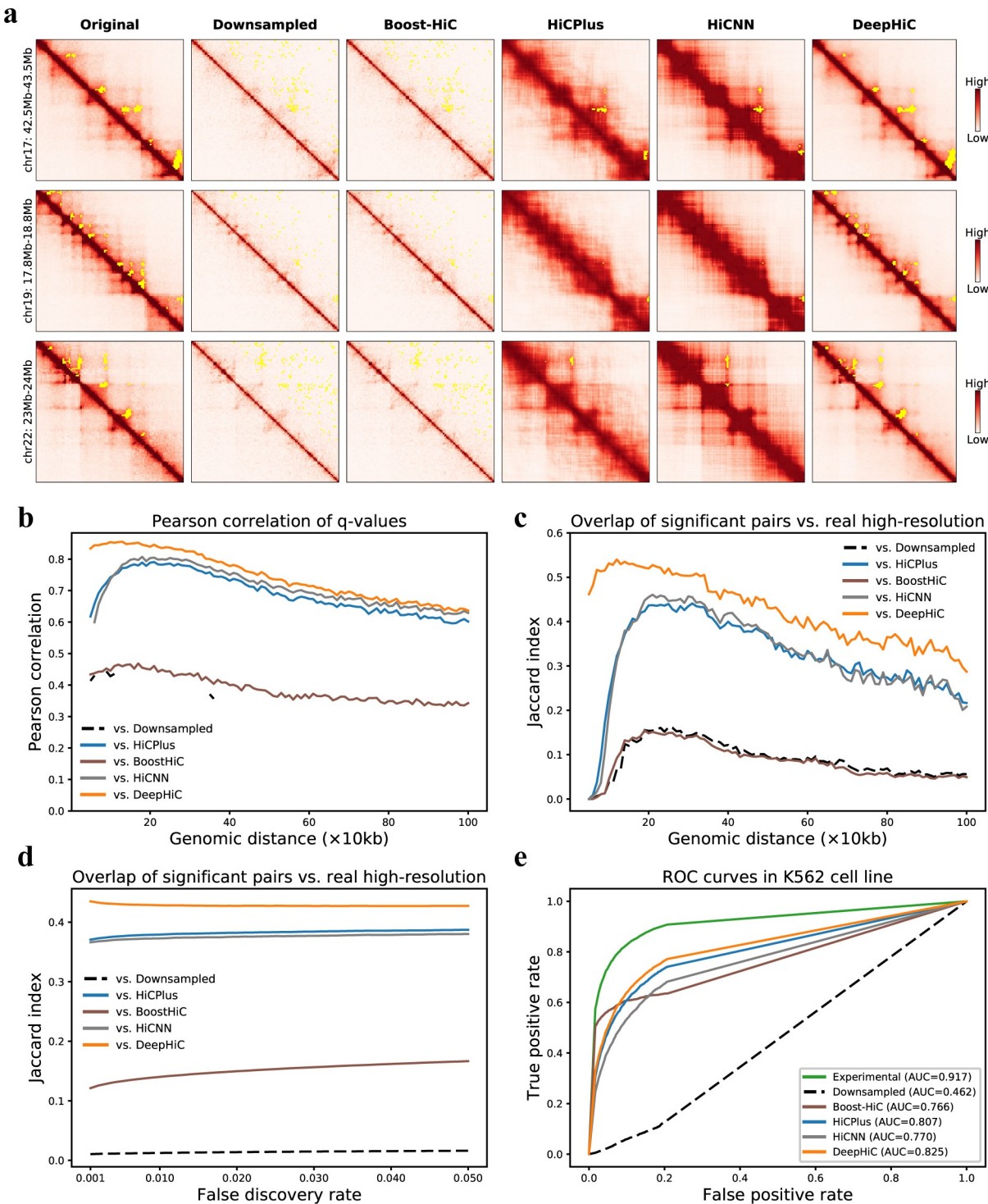

**Fig 4. Analyses of significant chromatin interactions identified by Fit-Hi-C software. (a)** Three representative sub-regions (1 Mb × 1 Mb) from chromosomes 17 and 22 (GM12878 cell line), with significant loci-pairs (cut-off is the 0.5 percentile of q-values) being marked with yellow points in the upper triangle of the heatmaps. **(b)** All q-values were treated as significance matrices. The Pearson correlations of q-values for non-experimental data vs. real Hi-C data at various genomic distances are presented. Missing values are *NaN* values derived by python (numpy). **(c)** We evaluated the overlap of significant loci-pair with real Hi-C data at each distance, using the preset cut-off. **(d)** We evaluated the overlap of all significant loci-pairs with various cut-off values, with respect to the false discovery rate which ranges from 0.001 to 0.05. **(e)** ROC analysis of overlap between interactions from CTCF ChIA-PET with identified interacting peaks from real high-resolution, downsampled, HiCPlus/HiCNN-enhanced, and DeepHiC-enhanced Hi-C matrices in the K562 cell line.

results (average AUC = 0.825). We also observed that the AUC score for the DeepHiC-enhanced matrix was significantly higher than both the AUC derived from the HiCPlus/HiCNN-enhanced matrix (p-value = 0, paired t-test) as well as the AUC derived from the downsampled matrix (p-value = 0, paired t-test).

## DeepHiC is more precise in detecting TAD boundaries

The detection of TADs is not as sensitive to resolution decline as algorithms for detecting TADs, we obtained roughly the same results when using the Hi-C data with various downsampling ratios [23]. However, we found that some refined TAD structures were shifted-even wrongly detected-in low-resolution data. Therefore, we continually assessed the performance of DeepHiC in recovering TADs, especially in fine-scale TADs. We calculated the $\Delta$ score of insulation scores across the entire genome for all four datasets (Methods). The zero-points within monotonic rising intervals are considered to be TAD boundaries. Fig 5A illustrates the insulation $\Delta$ scores derived from experimental high-resolution, downsampled, HiCPlus/BoostHiC/HiCNN-enhanced, and DeepHiC-enhanced Hi-C matrices, on chromosome 22, in the region between 20–22.7 Mb, from the *GM12878* dataset. The trends seemed similar, but enlarged views around the zero-points revealed that DeepHiC obtained the closest location of zero-points, while downsampled Hi-C and HiCPlus-enhanced Hi-C had bias of 20–50 kb. The Pearson correlation coefficients between $\Delta$ scores derived from experimental Hi-C and those derived from non-experimental Hi-C were 0.937, 0.953, and 0.992 for downsampled, HiCPlus-enhanced, and DeepHiC-enhanced data, respectively.

As for the two segmentations formed by TAD boundaries, we calculated all split points' distances and all intervals' overlap with another segmentation (see Methods), then investigated the properties of the resulting arrays. As shown in Fig 5B, we illustrated the distribution of all boundaries' distances from $S_{down}$, $S_{boosthic}$, $S_{hicplus}$, $S_{hicnn}$, and $S_{deephic}$ to $S_{origin}$ in the *GM12878* dataset via box plot. Boundary segmentations were derived from corresponding data. The distances of DeepHiC-enhanced data were significantly smaller than those of Boost-HiC-enhanced data (p-value = $1.4 \times 10^{-40}$, Wilcoxon rank-sum test), those of HiCPlus-enhanced data (p-value = $7.1 \times 10^{-14}$, Wilcoxon rank-sum test), those of HiCNN-enhanced data (p-value = 0.035, Wilcoxon rank-sum test) and those of downsampled data (p-value = $1.3 \times 10^{-193}$, Wilcoxon rank-sum test). We also investigated the distribution of the overlap of segmentations vs. experimental high-resolution data (Fig 5C). The results showed that our model had a high proportion of high *JI* (p-value $< 1 \times 10^{-20}$ for downsampled/BoostHiC-enhanced/HiCPlus-enhanced data, $< 0.001$ for HiCNN-enhanced data, Mann Whitney U-test), which indicates that more TADs are precisely matched with those in real Hi-C data. Same results of comparisons for other cell types are illustrated in S22 Fig.

## DeepHiC enhances prediction of chromatin loops in mouse early embryonic developmental stages

DeepHiC can be used to enhance the resolution of existing time-resolved Hi-C data obtained through early embryonic growth. These data are prone to low resolution due to limited cell population (40-kb in [40]). Therefore, algorithms for detecting significant interactions, when applied to these data, may produce results with a relatively high false positive rate. We demonstrate that DeepHiC can be applied to Hi-C data of mouse early embryonic development to enable identification of significant chromatin interactions with a considerably lower false positive rate. We applied Fit-Hi-C to both original low-resolution Hi-C contact matrices and DeepHiC-enhanced contact matrices (Fig 6A) and kept pairs of loci with q-values lower than a preset cut-off (0.5 percentile) as significant interactions (predicted loops). Chromatin loops

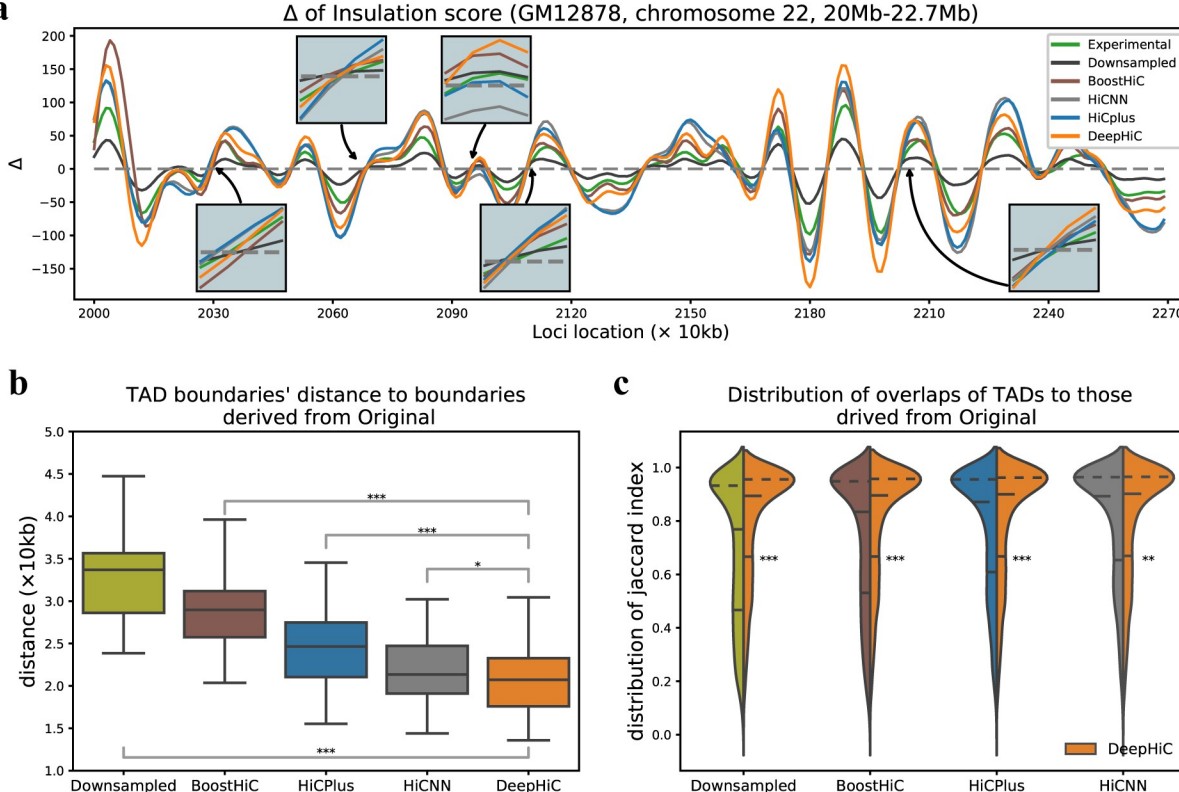

**Fig 5. Enhancements of DeepHiC in detecting TAD boundaries, using insulation score algorithm. (a)** Graphs of insulation Δ scores derived from different Hi-C data. TAD boundaries are zero-points of insulation Δ scores in ascending intervals. Enlarged photos show that zero-points derived from DeepHiC-enhanced data are closest to those derived from real high-resolution data. **(b)** Distances from TAD boundaries obtained from downsampled/enhanced data to those obtained from real high-resolution data. Boxplots show that distances of DeepHiC-enhanced data are significantly smaller than others (***: p-value < $1\times10^{-20}$, *: p-value < 0.05,Wilcoxon rank-sum test). The whiskers are 5 and 95 percentiles. **(c)** The distribution of the overlaps between TADs in downsampled/enhanced data and those in real high-resolution data. Higher proportion of high Jaccard indices (y-axis) was obtained with use of DeepHiC-enhanced data. ***: p-value < $1\times10^{-20}$, **: p-value < 0.001, Mann Whitney U-test. Dash lines in violin plots are quantiles.

regulate spatial enhancer-promoter contacts and are relevant to domain formation [4, 41], and anchors of Fit-Hi-C predicted significant interactions co-localize with open chromatin regions including insulators, enhancers, and promoters. In deeply sequenced Hi-C data of GM12878 cell line, significant interactions identified by Fit-Hi-C are significantly enriched in gene promoter and open chromatin regions compared to shuffled control (S23 Fig). Therefore, we evaluate the similarity of Fit-Hi-C significant interactions identified on mouse embryonic development Hi-C data to those identified in high-resolution Hi-C data according to the fraction of all significant interactions that connect promoter regions, as well as by the fraction connecting two accessible chromatin regions marked by ATAC-seq peaks. As shown in Fig 6B, significant interactions identified using DeepHiC enhanced Hi-C data are more likely to anchor at gene promoters than those identified using original Hi-C data. They are also more likely to co-localize with open chromatin regions at both of their anchoring loci than those predicted with original Hi-C data (Fig 6C). We mainly focused on the 8-cell stage and beyond because Hi-C data from earlier stages only demonstrate weak TADs and depleted distal chromatin interactions [40]. To generate control datasets, we randomly repositioned all predicted significant interactions for original Hi-C data, while maintaining the distance between anchors of each significant interaction, using the "shuffle" command in Bedtools [42]. We repeated this

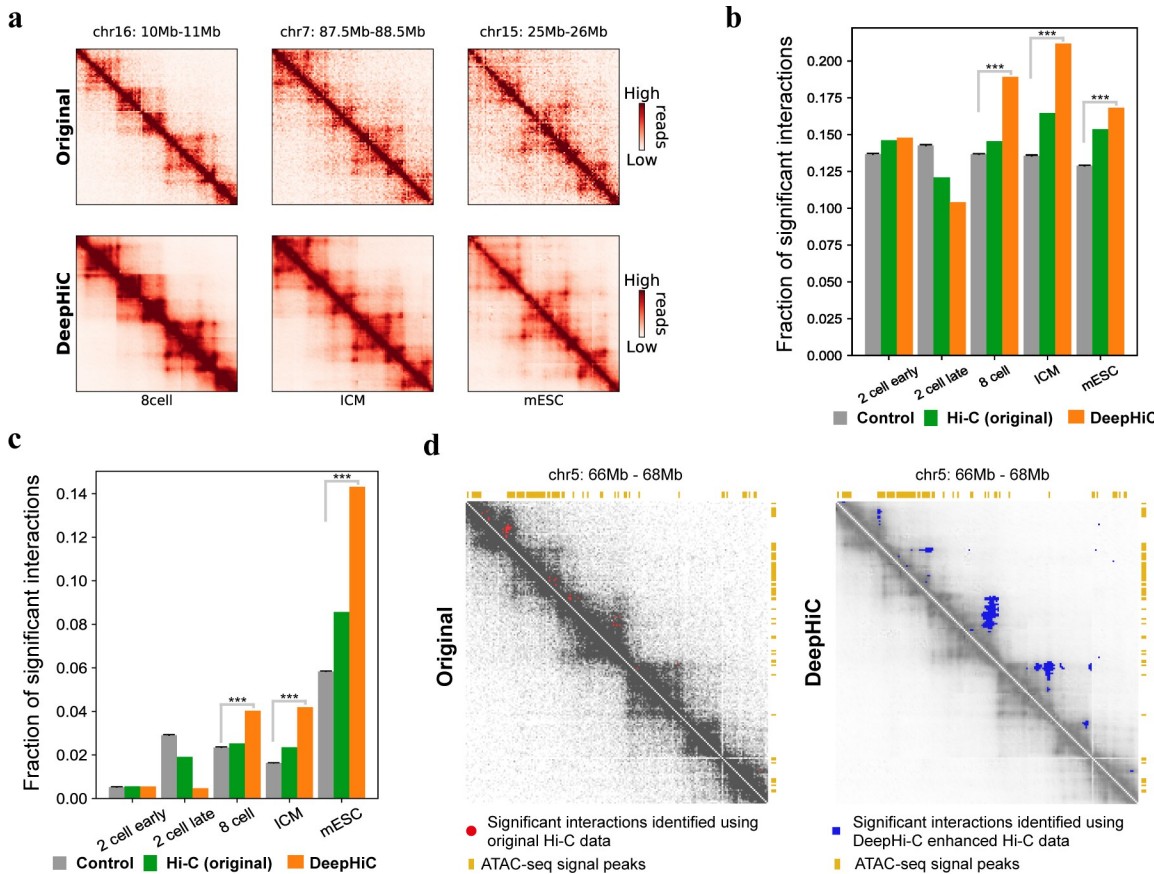

**Fig 6. Analysis of significant interactions identified using DeepHiC-enhanced Hi-C data of mouse early embryonic development.** **(a)** Heatmaps showing examples of original and DeepHiC enhanced contact matrices for various stage of embryonic development. **(b)** Fraction of significant interactions for which anchor loci intersected with gene promoters. Error bar: standard deviation. Significance: ***: p-value $< 1 \times 10^{-20}$ one-sample t-test. **(c)** Fraction of significant interactions for which both connected loci contain ATAC-seq signal peaks. Error bar: standard deviation. Significance: ***: p-value $< 1 \times 10^{-20}$, one-sample t-test. **(d)** A representative Hi-C contact matrix, with significant interactions as depicted for the 8-cell stage. Left panel: Original Hi-C contact matrix and predicted significant interactions (bold pixels inside red circles). Right panel: DeepHiC enhanced contact matrix and predicted significant interactions (blue pixels).

process 20 times to generate 20 random significant interaction datasets. We found that the fraction of predicted significant interactions that connected accessible loci was significantly higher for DeepHiC-enhanced Hi-C data, compared with random control data. Using an example at chromosome 5, we showed that significant interactions predicted using original Hi-C data were highly separated (Fig 6D). This is inconsistent with the known characteristics of significant interactions, as they are mostly located within TADs and are frequently observed as strong apexes of TADs and sub-TADs [4, 43]. Fig 6C shows that significant interactions as predicted using DeepHiC-enhanced Hi-C data are predominantly located within TADs, and at the apexes of TADs, where they co-localize with open chromatin regions. Therefore, DeepHiC is a powerful tool for studying chromatin structure during mammalian early embryonic development.

## Discussion

Hi-C is commonly used to map 3D chromatin organization across the genome. Since its introduction in 2009, this method has been updated many times in order to improve its accuracy

and resolution. However, owing to the high cost of sequencing, most available Hi-C datasets have relatively low resolution (40-kb to 1-Mb). The low-resolution representation of Hi-C data limits its application in studies of genomic regulatory networks or disease mechanisms, which require robust, high-resolution 3D genomic data.

In this study, we proposed a deep learning method, DeepHiC, for predicting experimentally-realistic high-resolution data from low-resolution samples. Our approach can produce estimates of experimental high-resolution Hi-C data with high similarity, using 1% sequencing reads. DeepHiC is built on state-of-the-art techniques from the deep learning discipline, including the GAN framework, residual learning, and perceptual loss. With using of the GAN framework, carefully designed net architecture, and loss functions in DeepHiC, it becomes possible to predict high-resolution Hi-C with high structural similarity of 0.9 to real high-resolution Hi-C. This approach may be used to accurately predict chromatin interactions, even in fine detail. Because of the huge quantity of parameters (~121,000) included in the network, DeepHiC may be used to approximate the real data, and to make predictions in other cell or tissue types. More importantly, enhancements afforded by DeepHiC favor the identification of significant chromatin interactions and TADs in Hi-C data. Finally, we also applied DeepHiC to Hi-C data pertaining to mouse early embryonic developmental stages, for which only low-coverage sequencing data were available, and enhancements afforded by DeepHiC facilitated identification of significant chromatin contacts for these data.

DeepHiC provides a GAN-based framework with which to enhance Hi-C data, and even other omics data. The GAN framework is a state-of-the-art technique in the deep learning field in recent years. The idea of adversarial training allows the deep model to capture learnable patterns efficiently and stably. DeepHiC is trained with real high-resolution data as target and is therefore a supervised learning paradigm. The quality of the target determines the upper-bound efficiency of the model. Here we used the most deeply sequenced GM12878 cell line data as a training set. It would be possible to retrain or fine-tune the model if more accurate Hi-C data were available, potentially reaching restriction-fragment resolution. Besides, we also performed a quasi-autoencoder training by taking low-coverage data to be both input and target, like an auto encoder, for ensuring that our model improves the sequencing depth rather than simply cleaning the data (S24 Fig). DeepHiC could be used not only to enhance existing low-resolution Hi-C data but also to reduce the experimental cost of sequencing in future Hi-C assays. As our method outperforms baseline methods, current low coverage Hi-C data could benefit from this improvement in performance. For example, Hi-C data imputed by our method can be used to identify significant interactions and TADs more similar to those identified with deeply sequenced Hi-C data. In some circumstances, in which the limitation on the number of cells stands in the way of producing high resolution Hi-C data, our method could provide an alternative solution to this problem. In addition, we develop a web-based tool (DeepHiC, http://sysomics.com/deephic) that allows researchers to enhance their own Hi-C data with just a few clicks. And the enhancement procedure runs in 3–5 minutes using single CPU (for example, enhancement on chromosome 1 of human will cost 4.7 minutes using a Xeon CPU E5-2682 v4 @ 2.5GHz). It will be faster when using a GPU (22s for Nvidia 1080ti). We trained several models based on various downsampled data. Translating the downsampling ratios to read coverage or data distribution is indispensable for users. We discuss the strategy of choosing between models in S2 Note. A caveat is that the low-coverage Hi-C data of input should have more than 10% non-zero entries.

In conclusion, DeepHiC introduced the GAN framework for enhancing the resolution of Hi-C interaction matrices. By utilizing the GAN framework and other techniques such as residual learning, DeepHiC can generate high-resolution Hi-C data using a low fraction of the original number of sequencing reads. DeepHiC can easily be used in a number of Hi-C data

analysis pipelines, and prediction could be executed quickly in minutes on the human genome.

## Materials and methods

### Hi-C data sources and processing

The high-resolution (10-kb) Hi-C data used for training and evaluating were obtained from GEO (https://www.ncbi.nlm.nih.gov/geo/) under accession number GSE63525. The primary Hi-C data in GM12878 cell line (HIC001-018) is denoted as *GM12878* dataset in this paper, and the corresponding replicate (HIC019-029) is denoted as *GM12878R* dataset for convenience. The high-resolution Hi-C contact maps for each dataset were derived from reads with mapping quality $> 30$. And we used the KR-normalization scheme [44] for normalized data.

Corresponding low-resolution data were simulated by randomly downsampling the sequencing reads to different ratios range from 1:10 to 1:100 (i.e., 1% reads). Downsampled data would typically be processed at lower resolution because of the shallower sequencing depths. In our experiments, low-resolution contact maps were built using the same bin size as used for high-resolution Hi-C to fit the models' requirement. All resolution enhancing methods compared in our study used this same procedure as reported in HiCPlus [17] to ensure fair comparisons.

Hi-C data pertaining to mouse embryonic development were obtained from GEO under accession number GSE82185. Hi-C matrices of 10-kb bin size were created using the HOMER (http://homer.ucsd.edu/homer/) analyzeHiC command with the following parameters: -res 10000 –window 10000.

ChIA-PET data for the CTCF target in the K562 cell line was obtained from ENCODE (https://encodeproject.org) under accession number ENCSR000CAC (file: ENCFF001THV. bed.gz). Chromosome regions were mapped to Hi-C bins with which they overlapped in ROC analysis. ATAC-seq data on mouse early embryonic development was obtained from GEO under accession number GSE66390. Data of DNase-seq peaks on GM12878 cell line was obtained from ENCODE under accession number ENCSR000EMT.

For Hi-C matrices in training, outliers are set to the allowed maximum by setting the threshold be the 99.9-th percentile. For example, 255 is about the average of 99.9-th percentiles for 10-kb Hi-C data, so all values greater than 255 are set to 255 for 10-kb Hi-C data. Then all Hi-C matrices are rescaled to values ranging from 0 to 1 by min-max normalization [45] to ensure the training stability and efficiency. Besides, cutoff values for downsampled inputs of our model were 125, 100, 80, 50, and 25 for 1/10, 1/16, 1/25, 1/50, and 1/100 downsampled ratios.

### DeepHiC architecture

In general, DeepHiC is a GAN model that comprises a generative network called *generator* and a discriminative network called *discriminator*. The *generator* tries to generate enhanced outputs that approximate real high-resolution data from low-resolution data, while the *discriminator* tries to tell generated data apart from real high-resolution data and reports the difference to the *generator*. The contest (hence "adversarial") between *generator* and *discriminator* promotes the *generator* learns to map from conditional input to a data distribution of interest.

As depicted in S1 Fig, the *generator* net ($G$) is a convolutional residual network (first row), while the *discriminator* net ($D$) is a convolutional neural network (second row). The $G$ net takes low-resolution matrices ($X$) as input and outputs enhanced matrices ($\hat{Y}$) with identical size. The adversarial component, the $D$ net, takes the enhanced output $\hat{Y}$ and the real high-

resolution data ($Y$) as input and outputs 0–1 labels. The green arrowed lines describe how data are processed in DeepHiC. The $G$ net, employs two layers: the convolutional layer (blue block) and the batch normalization (BN) layer [46] (yellow block). Together with elementwise sum operation (green ball) and skip-connection operation (green polyline), some of these layers form the residual blocks (ResBlocks) [47]. There are five successive ResBlocks in $G$. As for the activation function (pink block), we elected to use the Swish function [48] instead of the Rectified Linear Unit (ReLU) for activating some layers. The Swish function is defined as:

$$f(x) = x \cdot \sigma(\beta x),$$

where $\beta = 1$ and $\sigma$ is the sigmoid function. Swish has been shown to works better than ReLU in deep models [49]. Note that the final outputs of $G$ are scaled by:

$$g(x) = \frac{\tanh(x) + 1}{2}.$$

Thus, elements in output matrices range from 0 to 1. In general, the $G$ net contains about 121,000 parameters. The $D$ network is a convolutional network similar with the VGG network [50]. The number of kernels in a convolutional layer is depicted via block width: the more kernels, the wider the width of the block. The final output of $D$ is a scalar value ranges from 0 to 1 by a sigmoid function. More details of the hyperparameters of network architectures, such as kernel size and filter numbers, are summarized in S1 Table and S2 Table.

To establish the GAN paradigm for training (Fig 1A), we employed both the *generator* net $G$ and the *discriminator* net $D$. The $G$ net aims to generate enhanced outputs by approximating to the real high-resolution matrices $Y$, while the $D$ net attempts to distinguish the real $Y$ from the generated $\hat{Y}$. In the $D$ net, the value of output $\hat{y} = D(\hat{Y})$ is considered to be the probability of $\hat{Y}$ to be real data. Divergences between $\hat{Y}$ and $Y$, as well as the probability of $\hat{Y}$ to be real data, are minimized according to a carefully designed loss function. Besides, these two networks are trained alternatively by the backpropagation algorithm.

## Loss functions in DeepHiC

A critical point when designing a deep learning model is the definition of the loss function. Many methods have recently been proposed to stabilize training [51, 52] and improve the quality of synthesized images [37] by the GAN model. For DeepHiC, the binary cross entropy loss function for the $D$ network was used to measure the error of output, as compared with the assigned labels. Because real and generated high-resolution data are paired in practice, it can be described as:

$$L_D = \frac{1}{N} \sum_i \log(\hat{y}_i) + \log(1 - y_i),$$

where $i$ is the index for pairs of real and generated data, and $N$ is the number of pairs. Here we used $y = D(Y)$ and $\hat{y} = D(\hat{Y})$.

For *generator* loss, we used four loss functions, which were added to yield a final objective function. Firstly, we used MSE to measure the pixel-wise error between predicted Hi-C matrices and real high-resolution matrices, defined as:

$$MSE(\hat{Y}, Y) = \frac{1}{N} \sum_{i=1}^{N} (\hat{Y}_i - Y_i)^2,$$

which is also called L2 loss. The *MSE* loss function is broadly used for regression problems,

while the fact that *MSE* loss does not correlate well with the human perception of image quality [53] and overly smooths refined structures in images [27]. We also employ perceptual loss [37], however, based on the feature layers of the VGG16 network. We used total variation (TV) loss, derived from the total variation denoising technique, so as to suppress noise in images [54]. Final *generator* loss is yielded in combination with adversarial (AD) loss derived from *D* network and defined as:

$$L_G = l_{MSE} + \alpha \cdot l_{VGG} + \beta \cdot l_{TV} + \gamma \cdot l_{Ad}.$$

Note that $l_{Ad} = (\sum_i \hat{y}_i)/N$ without logarithmic transformation, which allows for fast and stable training of the *G* net [51]. Hyperparameters $\alpha,\beta,\gamma$ are scale weights that range from 0 to 1.

## Implementation of DeepHiC and performance evaluation

DeepHiC is implemented in Python scripts with PyTorch 1.0 [55]. After splitting *GM12878* dataset into a training set and a test set, the model was trained on the training set and tested on the test set during training process. The final model we used was trained on chromosomes 1–14. We divided contact matrices where the genomic distance between two loci is <2 Mb, as the average size of TAD is <1 Mb and there are few significance interactions outside TADs, thus could be omitted for training. The Adam optimizer [56] is used with a batch size of 64, and all networks are trained from scratch, with a learning rate of 0.0001. We trained the networks with 200 epochs. In order to yield loss terms on the same scale, the hyperparameters for *generator* loss were set as $\alpha = 0.006$, $\beta = 2 \times 10^{-8}$, and $\gamma = 0.001$. All training process were performed using an NVIDIA 1080ti GPU. A python code for model training and prediction is available at https://github.com/omegahh/DeepHiC.

In order to assess the efficiency of DeepHiC during training, we performed an improved measure called structure similarity index (SSIM) [57] to measure the structure similarity between different contact matrices. The SSIM score is calculated by sliding sub-windows between images. The measure for comparison of two identically sized sub-windows, *x* and *y* (from two images) is:

$$SSIM(x, y) = \frac{(2\mu_x \mu_y + C_1) \cdot (2\sigma_{xy} + C_2)}{(\mu_x^2 + \mu_y^2 + C_1) \cdot (\sigma_x^2 + \sigma_y^2 + C_2)},$$

where mean ($\mu$), variance ($\sigma$), and covariance ($\sigma_{xy}$) are computed with a Gaussian filter. They measure the differences of luminance, contrast, and structure between two images, respectively. $C_1$, $C_2$ are constants to stabilize the division with a weak denominator. In our experiments, the size of sub-windows and the variance value of Gaussian kernel are set as 11 and 3, respectively. And all compared matrices are rescaled by min-max normalization to same range to eliminate the differences of luminance in order to compare the contrast and structure differences.

## Dividing and reconstructing matrices

We divided the whole Hi-C contact maps into equal-sized square submatrices to be used as model inputs. It reduces the time and memory cost in batch training. The size of submatrices determines the features' dimension of each sample. Here we used the same size of 0.4 Mb × 0.4 Mb as described in HiCPlus, note that other choices such as 0.3Mb or 0.5Mb is also applicable in our workflow. So, each submatrix contains 40 × 40 = 1600 pixels at 10 kb resolution. As shown in Fig 1B, the intact low-resolution Hi-C matrix was divided into non-overlapping sub-

regions, then enhanced sub-regions were predicted from them (with outlier squashed and min-max normalization performed) by the *generator* network of DeepHiC. Finally, the high-resolution sub-matrices predicted were merged into a chromosome-wise Hi-C matrix, as the final enhanced output. Because our model is trained based on the contact maps where two bins < 2Mb genomic distance, we made the genome-wide predictions also on data where two bins < 2Mb.

### Identifying chromatin interactions and detecting TAD boundaries

Chromatin interactions [7] are identified using the commonly used software: Fit-Hi-C. We parallelized the software for faster running speed and suitable for our data. The modified code is available in https://github.com/omegahh/pFitHiC. Fit-Hi-C parameters were set as follows: *resolution* = 10*kb*, *lowerbound* = 2, *upperbound* = 120, *passes* = 2, *noOfBins* = 100. Significance was calculated only for intra-chromosome interactions. Since our model's output ranges from 0 to 1, we converted them to integer by multiplying 255 to be Fit-Hi-C inputs.

TADs were detected using the insulation score algorithm [14] with minor modifications: the width of the window used when calculating insulation score was set to 5 times of Hi-C matrix resolution to better detect the boundaries of finer-domain structures. We computed the delta score using insulation score of 5 nearest loci upstream and of 5 nearest loci downstream. We identified TADs as the genome region between center of 2 adjacent boundaries and regions containing low-coverage bins were excluded.

### Measurements for two TAD segmentations

We investigated the consistency of segmentations formed by different TAD boundaries in the genome. Here we calculated the distance of two segmentations and the corresponding overlap, defined as follows. We denote the two segmentations as $S$ and $T$, which are formulated in sets consisting of their split points:

$$S = \{s_1, s_2, \ldots, s_n\},$$
$$T = \{t_1, t_2, \ldots, t_m\},$$

where $m$, $n$ are numbers of split points. Thus, we could calculate the distance from one split point $s_i \in S$ to segmentation $T$, as follows:

$$d(s_i, T) = \min d(s_i, t_j), \quad \forall j = 1, 2, \ldots, m.$$

The overlap of an interval $I_S = (s_i, s_{i+1})$ from $S$, compared with T, could be measured as follows:

$$JI(I_S, T) = \max JI(I_S, I_T), \quad with \ I_T = (t_j, t_{j+1}), \quad \forall j = 1, 2, \ldots, m - 1,$$

### Implementation of baseline models

For baseline models, we only performed comparisons on data downsampled to 1/16 reads as they commonly used in their study [17, 25, 26]. The python source code for HiCPlus was obtained from https://github.com/zhangyan32/HiCPlus_pytorch, together with the codes for data processing and pre-trained model parameter file. We obtained HiCPlus results using the downloaded source code and pre-trained model parameter file. The scheme of data downsampling and reconstructing were implemented according to the description in its paper [17]. For Boost-HiC, the python source code was obtained from https://github.com/LeopoldC/Boost-HiC and implemented with *alpha* = 0.2. For HiCNN, we obtained its implementation code from http://dna.cs.miami.edu/HiCNN/ and pretrained model parameters from http://dna.cs.

miami.edu/HiCNN/checkpoint_files/. We used the "HiCNN_16" for experiments for 1/16 downsampled data.

## Supporting information

**S1 Note. The setting of colorbar for heatmaps plotting.**
(PDF)

**S2 Note. A guide of choosing trained model of DeepHiC.**
(PDF)

**S1 Fig. Overview of the architecture of DeepHiC model.** DeepHiC is a conditional generative adversarial network (cGAN) that contains two separated networks. The generator net (first row) takes low-resolution samples as input and generates the enhanced output. The discriminator net (second row), employed only during the training stage, discriminates real and generated high-resolution data. The objective function of the generator net is the sum of four loss functions: mean squared error (MSE) loss, perceptual loss, total variation (TV) loss, and adversarial (AD) loss, all of which are introduced according to the GAN paradigm. The settings used for convolution layers (blue blocks) are listed in detail in S1 Table and S2 Table.
(TIFF)

**S2 Fig. SSIM scores in the training process of different downsampled data.** the SSIM scores in test set increased gradually and converged during the training process.
(TIFF)

**S3 Fig. The peak-signal-to-noise ratio (PSNR) score in test set in the training process of different downsampled data.** The PSNR is derived from MSE, It equals $10 \times \log_{10}(1/MSE)$ in our experiment.
(TIFF)

**S4 Fig. Estimation of potential over-fitting in DeepHiC model.** To study the possible over-fitting issue in our model, we calculated the *generator loss* (*G* loss) during the training process on the training sets (chromosome 1–14) and test sets (chromosome 15–22) in GM12878 cell type. We observe that the loss in training and test sets keep the same trend in the entire training process. (test perform on 1/25 downsampling data)
(TIFF)

**S5 Fig. Cross validation of DeepHiC.** Training on different chromosomes in the GM12878 dataset. SSIM scores are evaluated in remaining chromosomes in GM12878 dataset besides training set. (test perform on 1/25 downsampling data)
(TIFF)

**S6 Fig. Training without GAN framework.** The SSIM scores in test set did not converge when training the generator net without pitting against the discriminator net.
(TIFF)

**S7 Fig. Training DeepHiC in IMR90 or K562 cell line data. a**) The SSIM scores in test set (chr15-chrX) are approaching nearly 0.9 when training DeepHiC on chr1-chr14 data in the K562 or IMR90 cell line. **b**) Pearson correlation in the same set (chr15-chrX in IMR90) when using trained model from different cell type.
(TIFF)

**S8 Fig. Performance in SSIM scores when predict from various downsampled data.** After training DeepHiC, we evaluate the performance of DeepHiC in enhancing low-resolution data

derived from different downsampled ratios. Comparing with the replicated assay in GM12878 cell line, DeepHiC achieved comparative performance in SSIM scores in both training and test set at each downsampling ratio.
(TIFF)

**S9 Fig. Performance evaluation in the test set when training 1/16 downsampled data.** We randomly downsampled the original reads to a 1:16 ratio as low-resolution input, then trained DeepHiC on chromosomes 1–14 and tested it on chromosomes 15–22 (i.e., test set), in GM12878 cell line. As training progressed, the structure similarity index (SSIM) between predicted and real high-resolution data gradually increased and converged on the summit value of 0.9.
(TIFF)

**S10 Fig. Enhancements of low-resolution data by DeepHiC in GM12878R.** Here we present the heatmap of three 1 Mb × 1 Mb (100 bins) sub-regions extracted from chromosomes 16, 17, and 22 from the replicate assay of GM12878 cell line. Colorbar setting: S1 Note.
(TIFF)

**S11 Fig. Enhancements of low-resolution data by DeepHiC in K562.** Here we present the heatmap of three 1 Mb × 1 Mb (100 bins) sub-regions extracted from chromosomes 16, 17, and 22 from the K562 cell line. Colorbar setting: S1 Note.
(TIFF)

**S12 Fig. Enhancements of low-resolution data by DeepHiC in IMR90.** Here we present the heatmap of three 1 Mb × 1 Mb (100 bins) sub-regions extracted from chromosomes 16, 17, and 22 from the MR90 cell line. Colorbar setting: S1 Note.
(TIFF)

**S13 Fig. Diagram of How SSIM score between two Hi-C matrices calculated.** In our experiments, we calculated SSIM of the 1Mb x 1Mb (100 bins x 100 bins) sub-regions at the diagonal as those regions cover the genome distance of interest. We calculate the mean of SSIMs between all 1Mb x 1Mb sub-regions with non-overlap at the diagonal across the entire genome to be the final SSIM score between two large Hi-C contact maps.
(TIFF)

**S14 Fig. Genome-wide comparative analysis in SSIM scores.** Between three types of non-experimental data: 1/16 downsampled, HiCPlus/Boost-HiC/HiCNN-enhanced, and DeepHiC-enhanced data in various cell types. We calculated the SSIM scores of three non-experimental data, as compare to real high-resolution data for all chromosomes.
(TIFF)

**S15 Fig. Genome-wide comparative analysis in correlation.** Between three types of non-experimental data: 1/16 downsampled, HiCPlus/Boost-HiC/HiCNN-enhanced, and DeepHiC-enhanced data in various cell types. We calculated the correlation three non-experimental data, as compare to real high-resolution data for all chromosomes at each genome distance.
(TIFF)

**S16 Fig. Profile along the genome of the first eigenvector of correlation maps derived from real high-resolution (experimental), DeepHiC-enhanced, and BoostHiC-enhanced matrices.** Quantitatively, correlation of DeepHiC and BoostHiC versus experimental are 0.952 and 0.966. And Jaccard indices are 0.
(TIFF)

**S17 Fig. Performance of DeepHiC in 1/25 and 1/36 downsampled data (trained in 1/16 downsampled data).** Correlations in GM12878 cell line when prediction 1/25 (50kb) and 1/36 (60kb) sequencing reads by using the trained model based on 1/16 sequencing reads.
(TIFF)

**S18 Fig.  Performance of DeepHiC in other cell lines' data from Rao et al., 2014 a, b)** The performances of DeepHiC in KBM7, NHEK, HUVEC, HMEC, and CH12-LX cell lines, down-sampling ratio is 1/16. **c)** Total read counts of 9 datasets (include 8 cell lines, the data of HeLa cell line is not available) from Rao et al., 2014. **d)** Correlations between Hi-C data across different cell types, black numbers represent correlations between two real data. Magenta numbers represent correlations between real data and predicted data (1/16 downsampled). Median correlation: median value of Pearson correlations from 50 kb to 1 Mb genomic distance
(TIFF)

**S19 Fig. Performance of DeepHiC in HI-C data prepared using 6-cutter enzyme.** Performances of correlation and SSIM scores when apply DeepHiC to GM12878 HindIII(&Ncol) Hi-C data (dilution Hi-C, HindIII&NcoI, HIC034-037).
(TIFF)

**S20 Fig. The number of significant loci-pairs in corresponding sub-regions in Fig 4A (denoted as I, II, and III).**
(TIFF)

**S21 Fig. Comparative analyses results of recovering loops in GM12878R/K562/IMR90 cell line data.** We investigated the correlation of significance matrices (first column), overlap of identified interactions with 0.5 percentile as cutoff (second column) at 100kb to 1Mb genomic distance in the *GM12878R*, *K562*, *IMR90* datasets, together with overlap of identified interactions with different cutoff according to false discovery rate from 0.001 to 0.05 (third column) in those three datasets.
(TIFF)

**S22 Fig. Comparative analyses results of recovering TADs in GM12878R/K562/IMR90 cell line data.** We investigated the performance in detecting TAD boundaries in *GM12878R* (first row), *K562* (second row) and *IMR90* (third row) datasets. ***: p-value $< 1x10^{-20}$, **: p-value $< 0.001$, Mann Whitney U-test.
(TIFF)

**S23 Fig. Properties of Fit-Hi-C identified significant interaction in high-resolution Hi-C data of GM12878 cell line. a)** Fraction of significant interactions of which both connected loci contain DNase-seq signal peaks. Error bar: standard deviation. **b)** Fraction of significant interactions anchor loci which are interested with gene promoters. Error bar: standard deviation.
(TIF)

**S24 Fig. Training DeepHiC using low-coverage input as both input and target, similar to an autoencoder. a)** When training DeepHiC using a quasi-autoencoder scheme, the trends of SSIM score in test set. **b)** Using different trained manners, the performance of correlations between predicted-GM12878R versus original GM12878 dataset in various Hi-C data.
(TIFF)

**S1 Table. Detailed settings for layers in generator network.**
(XLSX)

**S2 Table. Detailed settings for layers in discriminator network.**
(XLSX)

## Acknowledgments

We thank the Aiden Lab, the Wei Xie lab, and ENCODE Consortium for high-quality data.

## Author Contributions

**Conceptualization:** Cheng Li, Hebing Chen, Xiaochen Bo.

**Data curation:** Guifang Du, Cheng Quan, Chenghui Zhao, Ruijiang Li, Wanying Li, Hebing Chen.

**Formal analysis:** Hao Hong, Shuai Jiang, Hao Li, Hebing Chen.

**Funding acquisition:** Xiaochen Bo.

**Investigation:** Ruijiang Li.

**Methodology:** Hao Hong, Shuai Jiang, Hao Li.

**Project administration:** Cheng Li.

**Resources:** Xiaoyao Yin, Yangchen Huang.

**Software:** Hao Hong, Shuai Jiang, Cheng Quan, Chenghui Zhao, Ruijiang Li, Xiaoyao Yin, Yangchen Huang.

**Supervision:** Cheng Li, Hebing Chen, Xiaochen Bo.

**Validation:** Shuai Jiang, Hao Li, Guifang Du, Yu Sun, Huan Tao, Cheng Quan, Wanying Li.

**Visualization:** Hao Hong, Shuai Jiang.

**Writing – original draft:** Hao Hong, Shuai Jiang.

**Writing – review & editing:** Hao Hong, Shuai Jiang, Guifang Du.

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
