## [Decision Letter · Decision Letter 0]

29 Sep 2019

Dear Dr Bo,

Thank you very much for submitting your manuscript 'DeepHiC: A Generative Adversarial Network for Enhancing Hi-C Data Resolution' for review by PLOS Computational Biology. Your manuscript has been fully evaluated by the PLOS Computational Biology editorial team and in this case also by independent peer reviewers. The reviewers appreciated the attention to an important problem, but raised some substantial concerns about the manuscript as it currently stands. While your manuscript cannot be accepted in its present form, we are willing to consider a revised version in which the issues raised by the reviewers have been adequately addressed. We cannot, of course, promise publication at that time.

Sincerely,

Ferhat Ay, Ph.D

Associate Editor

PLOS Computational Biology

Weixiong Zhang

Deputy Editor

PLOS Computational Biology

[LINK]

Reviewer's Responses to Questions

**Comments to the Authors:**

Reviewer #1: The authors present a method, named DeepHiC, that aims to increase the effective sequencing depth of Hi-C experiments. The method relies on a pair of neural networks that are trained in an adversarial manner similar to image super-resolution. While the paper is very clearly written and easy to follow, there are several critical issues that should be addressed. Further, while the authors demonstrate in several contexts that DeepHiCs outperforms previous work, the manuscript may benefit from a further discussion of the ramifications of these improvements.

Major

- While the general problem of increasing the effective sequencing depth of Hi-C experiments is important, the specific formulation considered by the authors is problematic. Specifically, it will never be the case that one has high resolution data for a few chromosomes and needs to improve the quality of the remainder (the cross-chromosomal setting). Rather, a user would like a model that had been trained on high quality data from one cell type and could then be used to improve the quality of other cell types (the cross-cell type setting). This distinction is important because, in the cross-cell type setting, a strong baseline is simply using the high resolution contact map from the cell type used to train the model. The Hi-C+ paper addresses this setting explicitly.

- The authors should clarify what precisely constituted the ground truth labels. Specifically, was the Hi-C data that had outliers removed and then min-max scaled used solely for training, or was it also considered to be the ground truth? Considering these transformed values to be the ground truth would be unfair to the competing models that had been pre-trained on data sets which were processed in a different manner. Supplementary Note 1 suggests that the output from the models are not converted back to the original space because their ranges differ, with DeepHiC ranging from 0-1, and other methods ranging from 0-100. Further, the output from the webserver is between 0 and 1. It is okay for DeepHiC to be trained using data processed in a different manner than other approaches, but all approaches should be evaluated on the original Hi-C counts to ensure fairness. Additionally, because most outliers tend to occur on at low genomic distances, squashing these outliers only for DeepHiC may help explain why it outperforms the other models by so much at low genomic distances.

- It is unclear that performing better than an experimental replicate by such a large margin across all genomic distances is a reasonable result. The authors should comment on what such a result means and why there is such large variance between replicates. Generally one would think of an experimental replicate as almost an upper bound of performance. This result suggests that it is significantly better to use DeepHiC on a HiC map with low sequencing depth than to perform the actual high sequencing depth experiment. This may be the actual result, but it is a significant claim and should have more supporting evidence.

- When a researcher has a Hi-C map with moderate coverage, they will likely want to know by what factor they should improve the effective coverage of that map in order to discover various features of chromatin architecture. For example, should they use a model that improves coverage by 100x (by training using 1% of the reads) or only by 16x (by training using ~6% of the reads)? Is it always better to use a model that improves coverage the most, or does improvement plateau after a certain amount of boost? It would be useful to readers if the authors could provide guidance on which model is appropriate in which setting, preferably by showing the performance on downstream tasks of models trained to boost coverage by different amounts.

Minor

- The webserver did not work when tried

- An important distinction that should be investigated is whether the contact maps yielded by DeepHiC are better because they are effectively higher sequencing depth or because the generative network simply cleans the data. This question could be answered by training a model in the same manner as DeepHiC, but having the target be the same as the input, similar to an autoencoder. This model would demonstrate the performance one gets from cleaning the data. Thus, any improvement that DeepHiC has over it could reasonably be attributed to the importance of a higher sequencing depth.

- The authors clearly describe the architecture, hyperparameters, and composition of loss functions of the models that compose DeepHiC. However, this naturally leads to the question of how such were determined. While it is not necessary for the authors to perform grid-search using a validation set, they should at least comment on how such were determined and clarify that they were not determined based on performance on the test set.

- The SSIM score is difficult to interpret as the primary evaluation metric. While the authors soundly reason that using MSE as the objective function during training may yield overly smooth predictions, there is no reason not to use it to evaluate trained models, particularly because it is much more interpretable to most readers.

- It is unclear how predictions were made in a genome-wide fashion. Was a cross-validation approach used, as alluded to in the methods? This should be explicitly stated in the text and include the number of folds. Similarly, in the final results section, the data set used to train the DeepHiC model should be briefly described.

- The term "(q-value < 0.5-percentile)" is difficult to parse. Was it a q-value < 0.5 that was used, or were the bottom 0.5% of q-values used?

Reviewer #2: Review is uploaded as an attachment.

Reviewer #3: Hong, Jiang et al in the manuscript "DeepHiC: A Generative Adversarial Network for Enhancing Hi-C Data Resolution" implement a generative adversarial network that interpolates low resolution HiC maps to produce higher-resolution maps. Their implementation offers some quantitative improvements over previously published methods. Unfortunately the manuscript does not offer sufficient biological or methodological insight to justify publication in this journal. Their success in interpolating the data confirm the low amount of information entropy present in large HiC data sets, as it was previously shown. As a consequence, the manuscript does not seem of high importance to researchers in the field. I suggest submitting the manuscript to some more technical journal.

**Have all data underlying the figures and results presented in the manuscript been provided?**

Reviewer #1: Yes

Reviewer #2: Yes

Reviewer #3: Yes

PLOS authors have the option to publish the peer review history of their article (what does this mean?). If published, this will include your full peer review and any attached files.

Reviewer #1: No

Reviewer #2: Yes: Oana Ursu

Reviewer #3: No

---

## [Decision Letter · Decision Letter 1]

14 Jan 2020

Dear Dr Bo,

We are pleased to inform you that your manuscript 'DeepHiC: A Generative Adversarial Network for Enhancing Hi-C Data Resolution' has been provisionally accepted for publication in PLOS Computational Biology.

Before your manuscript can be formally accepted you will need to complete some formatting changes, which you will receive in a follow up email. Please be aware that it may take several days for you to receive this email; during this time no action is required by you. Once you have received these formatting requests, please note that your manuscript will not be scheduled for publication until you have made the required changes. Also you still need to address the minor suggestions from one of the reviewers and do a thorough grammatical check. 

In the meantime, please log into Editorial Manager at https://www.editorialmanager.com/pcompbiol/, click the "Update My Information" link at the top of the page, and update your user information to ensure an efficient production and billing process.

One of the goals of PLOS is to make science accessible to educators and the public. PLOS staff issue occasional press releases and make early versions of PLOS Computational Biology articles available to science writers and journalists. PLOS staff also collaborate with Communication and Public Information Offices and would be happy to work with the relevant people at your institution or funding agency. If your institution or funding agency is interested in promoting your findings, please ask them to coordinate their releases with PLOS (contact ploscompbiol@plos.org).

Thank you again for supporting Open Access publishing. We look forward to publishing your paper in PLOS Computational Biology.

Sincerely,

Ferhat Ay, Ph.D

Associate Editor

PLOS Computational Biology

Weixiong Zhang

Deputy Editor

PLOS Computational Biology

Reviewer's Responses to Questions

**Comments to the Authors:**

Reviewer #1: I am happy to report that the authors appear to have put a great deal of effort into their revision and have addressed all of my concerns. I would also like to apologize to the authors for my comment that their webserver did not work. The webserver did indeed work but took some time to run and I had forgotten to remove the comment after the webserver finished running.

Reviewer #2: Review is uploaded as an attachment.

**Have all data underlying the figures and results presented in the manuscript been provided?**

Reviewer #1: Yes

Reviewer #2: Yes

PLOS authors have the option to publish the peer review history of their article (what does this mean?). If published, this will include your full peer review and any attached files.

Reviewer #1: Yes: Jacob Schreiber

Reviewer #2: Yes: Oana Ursu

---

## [Editor Report · Acceptance letter]

13 Feb 2020

PCOMPBIOL-D-19-01246R1 

DeepHiC: A Generative Adversarial Network for Enhancing Hi-C Data Resolution

Dear Dr Bo,

I am pleased to inform you that your manuscript has been formally accepted for publication in PLOS Computational Biology. Your manuscript is now with our production department and you will be notified of the publication date in due course.

With kind regards,

Laura Mallard
